# Accelerating clinical development of a live attenuated vaccine against *Salmonella* Paratyphi A (VASP): study protocol for an observer-participant-blind randomised control trial of a novel oral vaccine using a human challenge model of *Salmonella* Paratyphi A infection in healthy adult volunteers

Naina McCann [1,2] Katherine Emary,[1,2] Nisha Singh,[1,2] Florence Mclean,[1,2] Susana Camara,[1,2] Elizabeth Jones,[1,2] Young Chan Kim,[1,2] Xinxue Liu [1,2] Melanie Greenland,[1,2] Kerry Conlin,[1,2] Jennifer Hill,[1,2] Marije Verheul,[1,2] Hannah Robinson,[1,2] Brian Angus [1,2] Maheshi N Ramasamy,[1,2] Myron M Levine,[3] Andrew J Pollard [1,2]

For numbered affiliations see end of article.

**Correspondence to**
Dr Naina McCann;
naina.mccann@paediatrics.ox.ac.uk

## ABSTRACT

**Introduction** This is the first efficacy study of an oral live attenuated vaccine against *Salmonella* Paratyphi A using a human challenge model of paratyphoid infection. *S.* Paratyphi A is responsible for 3.3 million cases of enteric fever every year, with over 19 000 deaths. Although improvements to sanitation and access to clean water are vital to reduce the burden of this condition, vaccination offers a cost-effective, medium-term solution. Efficacy trials of potential *S.* Paratyphi vaccine candidates in the field are unlikely to be feasible given the large number of participants required. Human challenge models therefore offer a unique, cost-effective solution to test efficacy of such vaccines.

**Methods and analysis** This is an observer-blind, randomised, placebo-controlled trial phase I/II of the oral live-attenuated vaccine against *S.* Paratyphi A, CVD 1902. Volunteers will be randomised 1:1 to receive two doses of CVD 1902 or placebo, 14 days apart. One month following second vaccination all volunteers will ingest *S.* Paratyphi A bacteria with a bicarbonate buffer solution. They will be reviewed daily in the following 14 days and diagnosed with paratyphoid infection if the predefined microbiological or clinical diagnostic criteria are met. All participants will be treated with antibiotics on diagnosis, or at day 14 postchallenge if not diagnosed. The vaccine efficacy will be determined by comparing the relative attack rate, that is, the proportion of those diagnosed with paratyphoid infection, in the vaccine and placebo groups.

**Ethics and dissemination** Ethical approval for this study has been obtained from the Berkshire Medical Research Ethics Committee (REC ref 21/SC/0330). The results will be disseminated via publication in a peer-reviewed journal and presentation at international conferences.

### STRENGTHS AND LIMITATIONS OF THIS STUDY

⇒ As a participant and observer blinded randomised control trial this study will provide high-quality evidence of the efficacy of an oral live attenuated *Salmonella* Paratyphi A vaccine candidate.

⇒ Immunology sampling throughout the trial will provide insight into the immune response following oral vaccination and infection with *S.* Paratyphi A which is so far poorly understood.

⇒ One limitation of this study is that the trial population of healthy UK adults may not be representative of the target population for a paratyphoid vaccine, namely children in endemic areas.

⇒ The model may under-estimate vaccine efficacy due to stringent diagnostic criteria.

**Trial registration number** ISRCTN15485902.

## INTRODUCTION
### Background

Enteric fever is a systemic febrile illness affecting 14.3 million individuals annually in 2017.[1] It is principally caused by *Salmonella enterica* serovars Typhi and Paratyphi A and is a common cause of morbidity, particularly in South and South East Asia. The proportion attributable to *Salmonella* Paratyphi A varies widely by geography but may be in the region of 25%,[1] and is increasing in some areas.[2–4]

In the UK, *S.* Paratyphi represents 40%–50% of travel-related enteric fever cases.[5–7]

Control of enteric fever in endemic areas is challenged by a lack of sensitive diagnostic tests, rising antimicrobial resistance and limited access to safe drinking water. Vaccines are likely to be a cost-efficient way to reduce disease burden. The recent implementation of the typhoid conjugate vaccine in endemic countries is likely to have a significant impact on reducing *S.* Typhi incidence[8–10] and it is unclear what impact wider typhoid vaccination coverage may have on the rates of *S.* Paratyphi A infection.[11] There are no current licensed vaccines against *S.* Paratyphi A.

Vaccine development for enteric fever has historically been hampered by the lack of an effective animal model to study the immunobiology of disease and the need for large sample sizes in efficacy trials. Controlled human infection models (CHIMs) offer a unique alternative to study immunological responses to and the efficacy of such vaccines.

The Oxford Vaccine Group (OVG), has developed an enteric fever CHIM where over 400 participants have safely undergone challenge since 2010.[12–15] In recent years the OVG has developed the first *S.* Paratyphi A human challenge model, offering an opportunity to test the efficacy and immunological response to novel *S.* Paratyphi A vaccine candidates.[15]

Several vaccines against *S.* Paratyphi A are in development, employing a range of approaches.[16 17] Oral live-attenuated vaccines offer several immunological and practical advantages to parenteral vaccines, including ease of administration and induction of a broad immunological response. The University of Maryland has developed such a vaccine, CVD 1902, for which a one-dose regimen has been shown to be safe and immunogenic in humans, although efficacy remains unknown.[5]

## STUDY AIMS AND OBJECTIVES

This study aims to assess the efficacy of the orally administered live-attenuated vaccine CVD 1902 and study the immune response both to *S.* Paratyphi A infection and vaccination. This manuscript is written in accordance with protocol version 3.1, dated 11 April 2022. Primary, secondary and exploratory objectives and outcomes are outlined in table 1.

## METHODS

### Study design and setting

This is an observer-blind, participant-blind, randomised, placebo-controlled trial of the oral live-attenuated vaccine CVD 1902 using a healthy adult participant CHIM of paratyphoid.

In total 74–76 participants will be randomised in a 1:1 ratio to receive a dose of not less than $2\times10^{10}$ CFU (Colony Forming Units) of CVD 1902 or placebo (sodium bicarbonate). As outlined in figure 1, all participants will receive two doses of vaccine or placebo 14 days apart. Twenty-eight days after their second vaccine or placebo dose participants will be challenged with *S.* Paratyphi A (strain NVGH308) at a dose of $1.5\times10^{3}$ CFU, the dose previously established to give a desired clinical/laboratory 'attack' rate of approximately 60%.[15]

The study will be conducted at the Centre for Clinical Vaccinology and Tropical Medicine, Oxford, UK, which is a fully equipped vaccine research site with available clinical inpatient facilities and a Category III level laboratory

| | Objectives | Outcome measures |
|---|---|---|
| **Table 1** Primary, secondary and exploratory objectives and outcomes of the Vaccine against *Salmonella* Paratyphi A study | | |
| Primary | To determine the relative protective effect of two doses of CVD 1902 given 14 days apart compared with placebo (sodium bicarbonate) in a healthy adult paratyphoid challenge model | The proportion of participants developing clinical or microbiologically proven paratyphoid infection following oral challenge with *S.* Paratyphi A in the group who have received two doses of CVD 1902 compared with those who have received two doses of placebo |
| Secondary | (a) To compare the clinical and laboratory features of the host responses following challenge with *Salmonella* Paratyphi A in participants vaccinated with CVD 1902 compared with placebo | Using clinical reporting, physical examination findings, microbiological and laboratory assays to compare the clinical course of paratyphoid infection after challenge between placebo and CVD 1902 groups |
| | (b) To compare the host immune response following vaccination with CVD 1902, compared with placebo including innate, antibody and cell-mediated responses and persistence of immunity and to relate these responses to the protective effect of vaccination | Immunological laboratory assays to assess innate, humoral, cell-mediated and mucosal responses to vaccination at baseline (day-42) and postvaccination time points |
| | (c) To compare the host immune response following *S.* Paratyphi A challenge following vaccination with CVD 1902 or placebo | Immunological laboratory assays to assess innate, humoral, cell-mediated and mucosal responses to challenge will be taken at various time points following challenge |
| | (d) To assess the safety and tolerability of CVD 1902 including faecal shedding | Clinical observation and participant recording of symptoms, both solicited and unsolicited plus safety laboratory data and microbiological data from blood and stool cultures following vaccination |
| | (e) To investigate immunological correlates of protection for *S.* Paratyphi A infection | Immunological response data postvaccination will be combined with vaccine efficacy data following *S.* Paratyphi A challenge to investigate if particular immunological markers could be used to predict protection from paratyphoid infection |

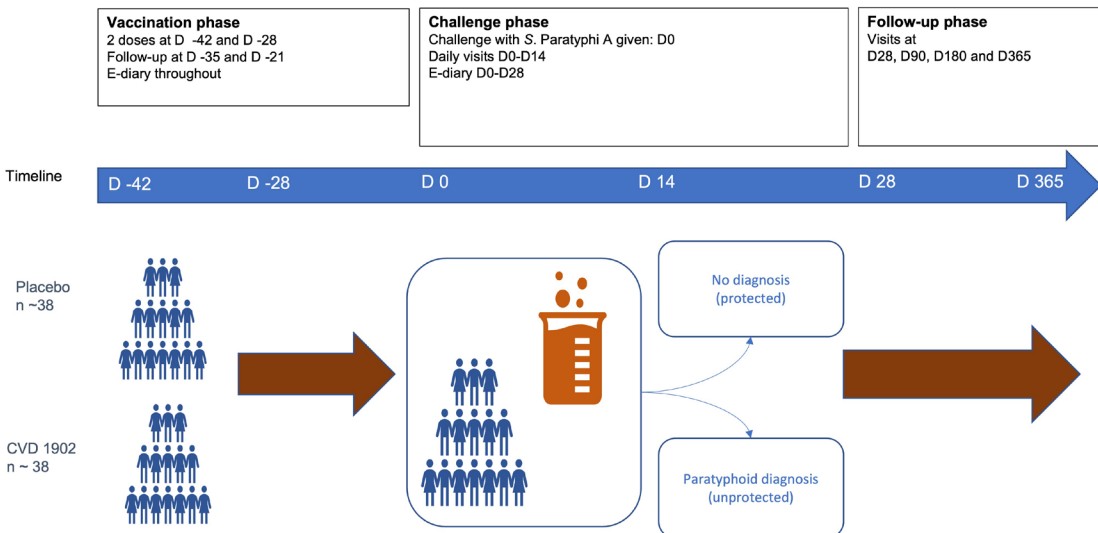

**Figure 1** Study design of the VASP trial investigating an oral vaccine against *S.* Paratyphi A. VASP, Vaccine against *Salmonella* Paratyphi A.

on-site. The UK is non-endemic for enteric fever; cases are typically travel related.[6]

### Recruitment and eligibility

All potential participants may be contacted by methods including but not limited to email, telephone, posters, leaflets, websites, advertisements in newspaper, radio and on social media and/or mail using an approved invitation letter or other approved advertising material to invite them to participate in the study. Participants will be reimbursed for their time, travel and inconvenience.

Male or female participants between 18 and 55 years inclusive who are in good health (as determined by medical history, physical examination and clinical judgement of study team and their general practitioner (GP)) and willing and able to provide written informed consent will be eligible for inclusion. Multiple other strict inclusion and exclusion criteria will be applied during the screening process to minimise the risk to participants and to minimise the risk of onward transmission to others. A summarised list of inclusion and exclusion criteria is provided in boxes 1 and 2 with a complete list provided in online supplemental material 1.

Participants will undergo a three-stage screening process to assess their suitability of the trial and to identify any of the inclusion and exclusion criteria. An online questionnaire will be followed by a telephone screening consultation and then a face-face consultation involving a physical examination, ECG, blood tests, urine dipstick, pregnancy test for females and gallbladder ultrasound examination.

Temporary exclusion criteria will also be applied on the days of vaccination and challenge, as outlined in online supplemental material 2.

### Trial procedures

Enrolment into the study will be day of first vaccination (D-42). Informed consent will be taken by an appropriately

trained clinician (online supplemental material 3). The detailed visit structure following enrolment can be found in online supplemental material 4.

### Randomisation

Participants will be randomised 1:1 to receive CVD 1902 or placebo on day-42. A statistician at OVG will generate a randomisation list using varying block sizes (2 and 4). The randomisation list aims to ensure equal target sample size is reached per group. The web-based randomisation system will ensure the allocation concealment is valid. Randomisation of participants will be carried out by unblinded study staff who are independent from the blinded team and do not perform any post vaccination procedures (such as ongoing eligibility, sample collection).

### Blinding

This study will be conducted observer-blind and participant-blind from the time of randomisation until participant unblinding which will occur once the *last* participant has completed their day 28 postchallenge visit. Observer and participant blinding is required to minimise the risk of bias on the reporting of adverse events (AEs) following the administration of vaccine and challenge. Unblinding may occur at an earlier time point in the event of participant withdrawal, or the occurrence of any serious AEs (SAEs) or reactions. This will be conducted under the guidance of the Data and Safety Monitoring Committee (DSMC).

### Vaccination visits

Vaccine or placebo will be administered at the D-42 visit and the D-28 visit. Participants must have fasted for 90 min prior to vaccination and pretreatment bicarbonate solution will be administered in the minute preceding vaccination to neutralise stomach acid. Participants will be directly observed for a minimum of 30 min following

---

### Box 1 Summary of inclusion criteria

**Inclusion criteria**

Participants must satisfy all the following criteria to be considered eligible for the study:

⇒ Willing and able to give informed consent for participation in the study.
⇒ Aged between 18 and 55 years inclusive at time of vaccination.
⇒ In good health as determined by medical history, physical examination and clinical judgement of the study team.
⇒ Willing to be available in Oxford for all required appointments.
⇒ Agree to comply with all study requirements, including capacity to adhere to good personal hygiene and infection control precautions.
⇒ Agree to allow study staff to contact his or her GP to access the participant's records and have their opinion solicited as to the participant's appropriateness for inclusion.
⇒ Agree to allow study staff to access NHS health records as required for study purposes.
⇒ Agree to allow his or her GP and UKHSA to be notified of participation in the study.
⇒ Agree to give his or her close household contacts written information informing them of the participants' involvement in the study and offering them voluntary screening for *Salmonella* Paratyphi A carriage.
⇒ Agree to have 24-hour contact with study staff during the 4 weeks postchallenge and are able to ensure that they are contactable by mobile phone for the duration of the vaccination and challenge period until antibiotic completion.
⇒ Have internet access to allow completion of the e-diary and real-time safety monitoring.
⇒ Agree to avoid antipyretic/anti-inflammatory treatment from challenge until advised by a study doctor or until 14 days after challenge.
⇒ Agree to refrain from donating blood for the duration of the study.
⇒ Agree to provide their National Insurance/Passport number for the purposes of TOPS registration and for payment.
⇒ Have received at least one dose of a SARS-CoV-2 vaccine ≥4 weeks prior to enrolment.
⇒ Agree to not receive other vaccinations during the 7 days before and after study vaccination and during the 7 days before or 21 days postchallenge.

GP, general practitioner; NHS, National Health Service; UKHSA, UK Health Security Agency.

---

### Box 2 Summary of exclusion criteria

**Exclusion criteria**

The participant will not be enrolled if any of the following apply:

⇒ History of significant organ/system disease that could interfere with trial conduct or completion.
⇒ Have any known or suspected impairment of immune function, alteration of immune function or prior immune exposure that may alter immune function to paratyphoid.
⇒ HLA-B27 positive.
⇒ Moderate or severe depression or anxiety.
⇒ Weight less than 50 kg.
⇒ Presence of implants or prosthetic material.
⇒ Anyone taking long-term medication that may affect symptom reporting or interpretation of the study results.
⇒ Contraindication to fluoroquinolones, macrolide antibiotics, cotrimoxazole or ceftriaxone.
⇒ Family history of aneurysmal disease.
⇒ Female participants who are pregnant, lactating or who are unwilling to ensure that they use effective contraception.
⇒ Full-time, part-time or voluntary occupations involving either direct contact with young children or highly susceptible patients or persons in whom typhoid infection would have particularly serious consequences, or commercial food handling.
⇒ Close household contact with young children or individuals who are immunocompromised (including pregnancy).
⇒ Scheduled elective surgery or other procedures requiring general anaesthesia during the study period.
⇒ Participants who have participated in another research study involving an investigational product that might affect risk of paratyphoid infection or compromise the integrity of the study within the 30 days prior to enrolment.

Detection of any abnormal results from screening investigations (at the clinical discretion of the study team).

Inability to comply with any of the study requirements.

Any other social, psychological or health issues which, in the opinion of the study staff, may

⇒ put the participant or their contacts at risk because of participation in the study,
⇒ adversely affect the interpretation of the primary endpoint data,
⇒ impair the participant's ability to participate in the study.

⇒ Have any history of allergy to vaccine/placebo components.
⇒ Having been resident in an enteric fever endemic country for 6 months or more.
⇒ Have previously been diagnosed with laboratory-confirmed typhoid or paratyphoid infection or been given a diagnosis compatible with enteric fever.
⇒ Have participated in previous typhoid or paratyphoid challenge studies.
⇒ Have received any oral typhoid vaccination at any time.
⇒ Have a prolonged corrected QT interval (>450 ms) on ECG screening.
⇒ Significant blood donation or planned blood donation prior to enrolment.

---

vaccination and then asked to complete an e-diary of their symptoms daily for 7 days following each vaccination (online supplemental material 5).

An in-person postvaccination review will occur 7 days following each vaccine where participants will be reviewed for any possible AEs and provide blood and stool samples.

### Challenge visits

*S.* Paratyphi A challenge will be administered by the oral route following pretreatment with sodium bicarbonate solution. Participants will be directly observed for 15 min following challenge and if they vomit within 60 min of challenge will be treated with antibiotics and withdrawn from the study.

Following challenge administration participants will be seen daily for 14 days and instructed to complete an online challenge e-diary to record oral temperatures two times a day and to describe any symptoms or the use of any medications for 28 days (online supplemental material 6).

Paratyphoid infection will be diagnosed based on certain prespecified criteria as seen in box 3.

---

**Box 3  Criteria for diagnosis of paratyphoid following human challenge**

**Paratyphoid fever is diagnosed if any of the following apply**
⇒ A positive blood culture for *Salmonella* Paratyphi A from 72 hours postchallenge.
⇒ A positive blood culture for *S.* Paratyphi A within 72 hours postchallenge, with one or more signs/symptoms of paratyphoid infection (such as recorded temperature ≥38.0°C).
⇒ Persistent positive blood cultures (two or more blood cultures taken at least 4 hours hours apart) for *S.* Paratyphi A within 72 hours postchallenge.
⇒ Oral temperature ≥38.0°C persisting for 12 hours in the 14 days following challenge.

Once diagnosed, participants are treated with antibiotics and monitored for severity of infection. Participants will be provided with medications for symptomatic control if required. At any stage, if clinically indicated, an additional review will be arranged by the clinical study team or if severely unwell they will be directed to the local accident and emergency department (or appropriate other medical facility) and relevant medical personnel from the trial and secondary care will be made aware. Antibiotics may also be started if felt to be clinically necessary or participant has severe symptoms not meeting diagnostic criteria.

### Follow-up visits
Participants will be followed up for 365 days following challenge. To detect chronic carriage of *S.* Paratyphi A and to confirm clearance, all participants are required to produce three stool samples obtained a minimum of 48 hours apart produced at least 1 week after completion of the antibiotic course. Participants with three successive negative stool samples will be considered to be fully treated for *S.* Paratyphi A infection and no longer an infection risk. The participants GP and local Health Protection Unit will be notified following challenge with *S.* Paratyphi A and then again once the participant has completed their clearance samples.

### Laboratory testing
Samples will be taken throughout the trial for safety, microbiological and immunological assays which are summarised in online supplemental material 7. Participants will be separately consented for their samples to be transferred to Oxford Vaccine Centre Biobank at the end of the study.

### Data management
#### Data collection
Data will be collected on case report forms (CRFs) and all the study data will be recorded directly into REDCap or onto a paper source document for later entry if direct entry is not available.

#### Data confidentiality
The study will comply with the UK General Data Protection Regulation and Data Protection Act 2018. Participant data will be deidentified other than for uses about which the participants will be specifically consented for. Any electronic databases and documents with participant identifying details will be stored securely and will only be accessible by study staff and authorised personnel.

Data collection and storage will be inspected throughout the study by the quality assurance team at OVG and monitoring will be carried out by the study Sponsor, University of Oxford Research Governance, Ethics and Assurance.

### Trial interventions
#### Vaccine
CVD 1902 is a live attenuated strain of *S.* Paratyphi A, an unlicensed, experimental oral vaccine for *S.* Paratyphi A infection developed by the Center for Vaccine Development at the University of Maryland School of Medicine in Baltimore. It contains two independently attenuating mutations, one in the *guaBA* locus, and the other in *clpPX*. *GuaBA* encodes genes involved in guanine biosynthesis and *clpPX* encodes a protease which targets the master flagellar regulator. The product is manufactured, tested and labelled according to current European Medicines Agency guidelines in keeping with Good Manufacturing Practice (GMP) by Bharat Biotech International Limited. The strain is fully sensitive to ciprofloxacin, trimethoprim/sulfamethoxazole and ampicillin.[18] This vaccine has been tested in adults in a phase I trial at the University of Maryland where a total of 30 healthy young adults received a single oral dose of CVD 1902 in a single-site, randomised, double-blinded phase I study.[18 19]

In this trial, vaccine recipients will receive two doses of CVD 1902 delivered in 30 mL carrier sodium bicarbonate solution, 14 days apart. Each dose will contain not less than $2\times10^{10}$ CFU, with an upper limit of $1.7\times10^{11}$ CFU.

The lower limit of the dose range for the vaccine in this trial corresponds to the highest dose administered in the previous phase I trial which was shown to be safe and immunogenic.[19] As higher doses were required in the phase I study to elicit immune responses, a two-dose schedule has been chosen to optimise immunogenicity while also testing a schedule that ultimately may be feasible at a population level. The rationale for using two doses of CVD 1902 14 days apart is to try and ensure an optimum immune response and protection as a previous study by Darton *et al* 2016 showed that a single-dose live attenuated oral vaccine for *Salmonella* Typhi M01ZH09 trialled within in a human challenge model did not provide protection, despite being immunogenic.[20]

#### Placebo
The vaccine placebo is 30 mL 1.3% wt/vol sodium bicarbonate solution made up using BP sodium bicarbonate powder with sterile water.

### Challenge

The *S.* Paratyphi A challenge strain, NVGH308, was isolated from a participant in a clinical study performed by the Oxford University Clinical Research Unit at Patan Hospital, Kathmandu, Nepal. It has been manufactured into batches to GMP standard by GenIbet BioPharmaceuticals, Portugal, and is supplied to OVG by Novartis Vaccines for Global Health. The strain is fully sensitive to ciprofloxacin, azithromycin, ampicillin and trimethoprim/sulfamethoxazole.

### Antibiotics

Antibiotics are used to treat diagnosed *S.* Paratyphi A infection or at day 14 postchallenge for those who have not been diagnosed.

The first line antibiotic will be oral ciprofloxacin 500 mg two times a day for 7 days, in-line with guidance on enteric fever treatment.[21] In fully susceptible *S.* Typhi isolates ciprofloxacin was shown to shorten time to resolution of symptoms, fever clearance time and bacteraemia when compared with azithromycin.[22] Participants will be given verbal and written information regarding the possible side effects of ciprofloxacin use. For any participant in whom a contraindication to these first line antibiotics becomes apparent oral trimethoprim/sulfamethoxazole, oral azithromycin or oral amoxicillin will be used as second-line therapy.

### Other trial interventions

Concomitant medication can be provided for symptomatic control of paratyphoid infection, before and after diagnosis. Concomitant medication after paratyphoid diagnosis includes antibiotics and antipyretics if required (online supplemental material 8).

### SAMPLE SIZE

Based on findings from the previous challenge studies performed at OVG, the assumption of attack rate of 58% was used in the control group.[14 15] To demonstrate a protective effect of 70% (ie, 30% relative risk in attacking rate), resulting in a reduction in attack rate from 58% in the control group to 17.4% in the vaccine group, 33 participants would be needed per group to achieve 90% power ($1-\beta$) at two-sided 5% significance level ($\alpha$).

We observed less than a 10% of dropout rate from previous vaccine trial using typhoid challenge model.[12] The sample size will be inflated to 37–38 participants per group to account for at least a 10% dropout. We expect to randomise 74–76 participants in total.

### STATISTICAL ANALYSIS

The analysis of the primary end point will be a calculation of the proportion of participants diagnosed with paratyphoid fever in the CVD 1902 group relative to the placebo group. The proportion of participants diagnosed with paratyphoid fever (ie, the attack rate) and the associated 95% CIs will be presented by group at day 28 after challenge. When calculating paratyphoid diagnosis proportions, the numerator will be the number of participants who meet the criteria for diagnosis and the denominator will be the per protocol population. The difference in proportions between the CVD 1902 and placebo groups will be analysed using Pearson's $\chi^2$ test (or Fisher's exact test if expected counts in any group are less than 5). To fulfil the primary objective, the protective effect of CVD 1902 over placebo will be calculated by:

$$PE = 100 \times (AR_{Placebo} - AR_{CVD\ 1902}) / AR_{Placebo} = 100 \times (1 - AR_{CVD\ 1902} / AR_{Placebo}),$$

Where PE is the protective effect and AR is the attack rate.

Time-to-event analyses of individual components of the primary outcome (positive blood culture and oral temperature $\geq 38.0°C$) will be conducted using the Kaplan-Meier method and will include all participants. Participants not meeting the criteria for an individual component of the primary endpoint will be censored in the analysis at the final monitoring for those undiagnosed. Participants who withdrew or had potential interference with vaccine effect or infection challenge (eg, treated prior to day 14 with no diagnosis of paratyphoid) will be censored in the analysis at the time of withdrawal or interference.

Immunogenicity data are expected to be highly skewed and will be log-transformed prior to analysis. Results will be presented as geometric means with 95% CIs. Values below the limit of detection (LLOD) will be imputed with half the value of the LLOD.

### ETHICS AND DISSEMINATION

### Ethical approval, regulation and governance

Ethical approval for this study has been obtained from the Berkshire Research Ethics Committee and Health Research Authority (REC reference: 21/SC/0330) and the study will run in accordance with the ethical principles of the Declaration of Helsinki.

The use of an unlicensed vaccine in this study, CVD 1902, is regulated by the Medicines for Human Use (Clinical Trials) Regulations 2004 (Medicines and Healthcare products Regulatory Agency (MHRA)). As CVD 1902 is a genetically modified organism, relevant approvals have been obtained for its use in this study from the Department of Environment, Food and Rural Affairs, Oxford University Hospitals NHS Foundation Trust and the Centre for Clinical Vaccinology and Tropical Medicine. The challenge agent falls outside the remit of these regulators and therefore is judged according to common law and best practice.[23] Regional ethics committees therefore provide support in governing challenge studies.

An independent DSMC consisting of an experienced group of infectious disease clinicians, scientists and a statistician will be appointed to provide real-time oversight of safety and trial conduct. The DSMC will review safety data collated from the electronic CRFs (eCRFs) and e-diaries

including solicited and unsolicited symptoms, laboratory results and vital signs. Following prime vaccination of a sentinel group of six participants relevant safety data will be reviewed by the DSMC. If there are no safety concerns these participants can proceed to their second vaccination and further participants can begin vaccination. Further DSMC reviews will occur regularly throughout the trial.

A development safety update report for the investigational medicinal product will be prepared annually, on the anniversary of the MHRA approval for the trial.

### Risks

The general risks to participants in this study are associated with potential side effects from the vaccine or placebo, symptomatic infection following challenge and the small risk of subsequent complications from paratyphoid infection. Additionally, there is a small risk of secondary transmission of *S.* Paratyphi A to close contacts of participants. The university has a specialist insurance policy in place, which would operate in the event of any participant suffering harm as a result of their involvement in the research.

### Vaccine

In the phase I trial 'CVD 1902 was well tolerated without clinically significant adverse reactions attributed to the vaccine'.[18] Hypersensitivity to the vaccine is extremely unlikely given the constituents present in the vaccine preparation but there may be unforeseeable side effects. A study doctor will check participants postvaccination symptoms daily in relation to the individual and trial stopping rules (online supplemental material 9).

### Challenge

Some study participants will develop symptomatic paratyphoid infection following challenge, for which risks include development of severe disease or complications. These risks will be greatly minimised by daily reviews by a clinical study team member, daily review of e-diary entries, regular safety bloods and early treatment with an effective antimicrobial. A previous challenge study using the same strain of *S.* Paratyphi A at OVG has demonstrated a good safety profile.[15]

### Carriage

The likelihood of developing chronic carriage is extremely low in the challenge setting. Participants are treated with ciprofloxacin, a fluoroquinolone antibiotic which is the preferred class of antibiotics for prevention of chronic carriage.[24] To ensure clearance of infection and to exclude chronic carriage, stool samples for culture will be obtained on completion of the initial antibiotic course. If participants remain positive after a second course of antibiotic treatment then participants will be referred to an Infectious Diseases Consultant at the Oxford University Hospitals NHS Foundation Trust for further management.

### Close contacts

In view of the low infectivity of *S.* Paratyphi A without bicarbonate buffer and the high standard of hygiene and sanitation in the UK, secondary transmission of the challenge strain to household or other close contacts after discharge is highly unlikely. It is thought that typhoidal Salmonellas, unlike *Shigella* sp, enterohaemorrhagic *Escherichia coli*, or hepatitis A virus, are virtually never transmitted by direct faecal–oral contact. This is in part due to the higher oral inoculum of these bacteria required to cause clinical disease.

It is acknowledged, however, that transmission within households can occur if the individual excreting *S.* Paratyphi A fails to practice effective hand washing after defecation and is subsequently involved in uncooked food preparation. Therefore, throughout the period of possible excretion of the challenge strain, participants must practice stringent hand washing techniques after defecation. Participants will be given soap and paper towels for use at home and detailed advice on how to prevent transmission of *S.* Paratyphi A. Participants will be taught and observed practising good hand hygiene technique at their initial challenge visit.

### AE reporting

All AEs occurring from enrolment to day 28 postchallenge will be recorded and graded (online supplemental material 10). Solicited AEs will be recorded and graded (0–4) by the participant in a vaccine e-diary for 7 days postvaccination and a challenge e-diary 21 days postchallenge. Solicited AEs will be reviewed daily during the periods of recording by a study clinician and if they have concerns about the severity or frequency of an event this will be followed up with the participant. Unsolicited AEs will be recorded by the participant in the e-diaries any time from first vaccine administration until day 28 postchallenge.

All solicited AEs recorded in the e-diary will be automatically assumed to be related to the vaccine or challenge (depending on whether recorded in vaccine diary or challenge diary, respectively) whereas all unsolicited AEs will undergo causality assessment by a clinician as to whether related to the vaccine, challenge agent or other medications.

Laboratory results and vital signs will be entered directly into an eCRF and automatically severity graded. Any clinically significant results will be reported as AEs and causality assessed.

Medically attended non-SAEs occurring between day 28 and day 90 elicited at the day 90 visit will not receive a causality assessment. All SAEs will be recorded from time of consent and reported to the DSMC within 24 hours of notification. SAEs will receive a causality assessment throughout the study, at the time of reporting. All SAEs at least possibly related to CVD 1902 will be considered unexpected and be reported to the MHRA and REC as serious unexpected serious adverse reactions within the regulatory timelines. AEs of special interest are listed in

---

**Box 4    Adverse events of special interest in the Vaccine against *Salmonella* Paratyphi A trial**

**Adverse events of special interest**
⇒ Severe paratyphoid infection.*
⇒ Failure to clinically or bacteriologically cure a participant of paratyphoid infection.
⇒ Progression to chronic carrier state.
⇒ Relapse of paratyphoid infection.
⇒ Transmission of *Salmonella* Paratyphi A to a contact of a participant.
⇒ AEs requiring a physician visit or emergency department visit which, in opinion of study staff, are related to challenge with *S*. Paratyphi A.
⇒ Pregnancy.

*Severe paratyphoid infection defined as illness that includes any of the following criteria: oral temperature >40°C, systolic blood pressure <85 mm Hg, significant lethargy or confusion, gastrointestinal bleeding, gastrointestinal perforation or any grade 4 or above laboratory abnormality.

box 4. These will be reported to the DSMC as soon as possible but within 7 days of discovery.

### Dissemination

Dissemination of the findings is planned for publication in peer-reviewed journals and at scientific international conferences.

### PATIENT AND PUBLIC INVOLVEMENT

The protocol, study information booklet and recruitment materials were reviewed by a local patient consultation group who provided feedback and comments on the initial documents. Their comments led to changes in the participant-facing documents, ensuring they are easy and clear for participants to understand.

### DISCUSSION

This study will be the first efficacy analysis of a live-attenuated oral vaccine against *S*. Paratyphi A using a human challenge model. Resulting data has the potential to be a critical step in the future licensure of the first vaccine against *S*. Paratyphi A. The role of an *S*. Paratyphi A vaccine is most likely to be part of a bivalent or even multivalent vaccine against both typhoidal and potentially non-typhoidal serovars. Deployment of a bivalent vaccine against *S*. Typhi and *S*. Paratyphi A would address the burden of enteric fever within an Asian context where both pathogens are endemic. In other areas such as sub-Saharan Africa, non-typhoidal serovars cause a significant burden of disease hence a multivalent vaccine incorporating typhoidal and non-typhoidal serovars may prove more cost-effective as a single approach regardless of geographical setting. However, in order to contribute to this vaccine development, an efficacious *S*. Paratyphi A vaccine is required. This trial will act as a proof of concept efficacy trial for such a vaccine, CVD 1902.

Reliable immunological correlates of protection for *S*. Typhi and *S*. Paratyphi A have not yet been identified.

Progress has been made in *S*. Typhi with data from prior *S*. Typhi challenge trials identifying anti-Vi IgA and IgG1 antibodies correlated with protection following Vi-TT and Vi-PS vaccination.[25 26] However, *S*. Paratyphi A lacks the Vi antigen and its immunobiology and pathogenesis is less well understood than it is for *S*. Typhi. The lipopolysaccharide O:2 has been identified as an important virulence factor for *S*. Paratyphi and the flagellar (H) antigens may also play a role in protection. During this trial we will evaluate both the humoral and cellular response to both CVD 1902 and *S*. Paratyphi challenge and in doing so provide invaluable immunological data and further our understanding of immunity to *S*. Paratyphi infection and correlates of protection.

As a CHIM the conditions during this trial will differ from endemic areas. The population in this trial is healthy UK adults whereas the majority of paratyphoid infections in endemic areas occur in young children. Although different physiologically, these groups may be similar from an immunological perspective as they are both paratyphoid naïve. The stringent diagnostic criteria in this trial may also underestimate vaccine efficacy by picking up subclinical bacteraemia, while patients in endemic areas are more likely to be diagnosed on symptoms alone. Nevertheless, the results of the typhoid conjugate vaccine human challenge study showed that this model can be used to provide early, representative data on vaccine efficacy that does translate to efficacy in field studies.[8 12]

Although no licensed vaccines against *S*. Paratyphi A exist there are several other mono and bivalent vaccines in development. Conjugate vaccines use the lipopolysaccharide O-antigen (O:2) of paratyphoid conjugated to a protein carrier, either to tetanus-toxoid or $CRM_{197}$ a nontoxic mutant of diptheria toxin. Phase I/II studies of O:2-TT showed this vaccine was safe and immunogenic and further phase II trials are currently underway[27] with a bivalent formation combined with Vi-TT also in development. Preclinical studies demonstrated $O:2-CRM_{197}$ was immunogenic when used alone or with $Vi-CRM_{197}$. Novel vaccines, such as nanoparticle-based vaccines and MAPS-based vaccines are also in development.[28]

As the first trial using a human challenge model to test efficacy of an *S*. Paratyphi vaccine, this trial paves the way for potential testing of other *S*. Paratyphi A vaccine candidates. Alongside improvements in sanitation and access to safe water supply, the development of an effective vaccine against *S*. Paratyphi A, has the potential to significantly impact on the burden of enteric fever in endemic countries.

**Author affiliations**
[1]Department of Paediatrics, Oxford Vaccine Group, University of Oxford, Oxford, UK
[2]NIHR Oxford Biomedical Research Centre, Oxford, UK
[3]Center for Vaccine Development and Global Health, University of Maryland School of Medicine, Baltimore, Maryland, USA

**Contributors** AJP, BA and MNR are the chief and principal investigators, who alongside MML, conceived and developed the study. NM and KE are research fellows who drafted the manuscript and protocol. XL, MG and KC are the study

statisticians. FM, SC, EJ, YCK, JH, MV contributed to the laboratory aspects of the study design. NS and HR helped in design of the final study protocol and coordinated regulatory approvals. All authors read and approved the final manuscript.

**Funding** This work was supported by Medical Research Council, grant number MR/R025347/1.

**Competing interests** AJP is chair of the UK Department of Health and Social Care's (DHSC) Joint Committee on Vaccination and Immunisation (JCVI), was a member of WHOs Strategic Advisory Group of Experts until 2022, receives consulting fees from Shionogi and is in receipt of grants from MRC, AstraZeneca, Gates Foundation, Wellcome Trust, NIHR and European Commission through the University of Oxford. MML is a coinvestigator on the MRC grant for this study, holds patents for Attenuated Salmonella Enterica Serovar Paratyphi A and Uses Thereof in the USA and UK, Germany, France, Italy and Spain, is a member of the FDA Vaccines and Related Biological Products Advisory Committee (VRBPAC) and a member of the NIH Data and Safety Monitoring Board (DSMB) to review and monitor US-government-supported clinical trials of candidate COVID-19 vaccines (2020 to present). MNR works as a principal investigator on clinical vaccine trials funded by AstraZeneca and Moderna but receives no personal financial payment for this work.

**Patient and public involvement** Patients and/or the public were involved in the design, or conduct, or reporting, or dissemination plans of this research. Refer to the Methods section for further details.

**Patient consent for publication** Not applicable.

**Provenance and peer review** Not commissioned; externally peer reviewed.

**ORCID iDs**
Naina McCann http://orcid.org/0000-0001-5864-1175
Xinxue Liu http://orcid.org/0000-0003-1107-0365
Brian Angus http://orcid.org/0000-0003-3598-7784
Andrew J Pollard http://orcid.org/0000-0001-7361-719X

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
