## [Reviewer comments · BMJ Open]

ARTICLE DETAILS

TITLE (PROVISIONAL)	Accelerating Clinical Development of a Live Attenuated Vaccine against Salmonella Paratyphi A (VASP): Study protocol for an observer-participant-blind randomised control trial of a novel oral vaccine using a human challenge model of Salmonella Paratyphi A infection in healthy adult volunteers
AUTHORS	McCann, Naina; Emary, Katherine; Singh, Nisha; Mclean, Florence; Camara, Susana; Jones, Elizabeth; Kim, Young Chan; Liu, Xinxue; Greenland, Melanie; Conlin, Kerry; Hill, Jennifer; Verheul, Marije; Robinson, Hannah; Angus, Brian; Ramasamy, Maheshi; Levine, Myron M; Pollard, Andrew

VERSION 1 – REVIEW

REVIEWER	Erdem, Rahsan PATH, CVIA
REVIEW RETURNED	17-Nov-2022

GENERAL COMMENTS	Very well written protocol, investigators defined many outcomes clearly except the follow up periods and parameters of solicited, unsolicited, AESI and SAEs are clearly defined in the outcome section, or in the body of the protocol. Suggest revising for further clarity.
--

REVIEWER	Khayeka-Wandabwa, Christopher D African Population and Health Research Center (APHRC), Nairobi, Kenya
REVIEW RETURNED	17-Feb-2023

GENERAL COMMENTS	The protocol is detailed and solid in flow with regard to pertinent information necessary for such kind of publications. I have one clarification question: Page 13 of 52 line 331-332: “The S. Paratyphi A challenge strain, NVGH308, was isolated from a participant in a clinical study 332 performed by the Oxford University Clinical Research Unit at Patan Hospital, Kathmandu, Nepal” Would the authors confirm that the due bioethics for transfer of research materials was followed in obtaining the sample from Nepal?
---

VERSION 1 – AUTHOR RESPONSE

Reviewer: 1

Dr. Rahsan Erdem, PATH

Comments to the Author:

Very well written protocol, investigators defined many outcomes clearly except the follow up periods and parameters of solicited, unsolicited, AESI and SAEs are clearly defined in the outcome section, or in the body of the protocol. Suggest revising for further clarity.

The adverse events section of the manuscript has been revised to include more detail on the periods and parameters of adverse events and supplementary material 10 has been added to show the full parameters of all adverse events.

Reviewer: 2

Mr. Christopher Khayeka-Wandabwa, African Population and Health Research Center (APHRC), Nairobi, Kenya

Comments to the Author:

The protocol is detailed and solid in flow with regard to pertinent information necessary for such kind of publications. I have one clarification question:

Page 13 of 52 line 331-332: “The S. Paratyphi A challenge strain, NVGH308, was isolated from a participant in a clinical study 332 performed by the Oxford University Clinical Research Unit at Patan Hospital, Kathmandu, Nepal”

Would the authors confirm that the due bioethics for transfer of research materials was followed in obtaining the sample from Nepal?

Ethical approval was sought and gained for this study, which includes collection and use of the S. Paratyphi A strain.